# Current Knowledge on Biomaterials for Orthopedic Applications Modified to Reduce Bacterial Adhesive Ability

**DOI:** 10.3390/antibiotics11040529

**Published:** 2022-04-15

**Authors:** Valeria Allizond, Sara Comini, Anna Maria Cuffini, Giuliana Banche

**Affiliations:** Bacteriology and Mycology Laboratory, Department of Public Health and Pediatric Sciences, University of Torino, Via Santena 9, 10126 Turin, Italy; valeria.allizond@unito.it (V.A.); annamaria.cuffini@unito.it (A.M.C.); giuliana.banche@unito.it (G.B.)

**Keywords:** prosthetic joint infections, septic loosening, multifunctional biomaterials, UHMWPE, titanium alloys, poly(ε-caprolactone), antibacterial properties, vitamin E, oxidation, silver, essential oils

## Abstract

A significant challenge in orthopedics is the design of biomaterial devices that are able to perform biological functions by substituting or repairing various tissues and controlling bone repair when required. This review presents an overview of the current state of our recent research into biomaterial modifications to reduce bacterial adhesive ability, compared with previous reviews and excellent research papers, but it is not intended to be exhaustive. In particular, we investigated biomaterials for replacement, such as metallic materials (titanium and titanium alloys) and polymers (ultra-high-molecular-weight polyethylene), and biomaterials for regeneration, such as poly(ε-caprolactone) and calcium phosphates as composites. Biomaterials have been designed, developed, and characterized to define surface/bulk features; they have also been subjected to bacterial adhesion assays to verify their potential capability to counteract infections. The addition of metal ions (e.g., silver), natural antimicrobial compounds (e.g., essential oils), or antioxidant agents (e.g., vitamin E) to different biomaterials conferred strong antibacterial properties and anti-adhesive features, improving their capability to counteract prosthetic joint infections and biofilm formation, which are important issues in orthopedic surgery. The complexity of biological materials is still far from being reached by materials science through the development of sophisticated biomaterials. However, close interdisciplinary work by materials scientists, engineers, microbiologists, chemists, physicists, and orthopedic surgeons is indeed necessary to modify the structures of biomaterials in order to achieve implant integration and tissue regeneration while avoiding microbial contamination.

## 1. Introduction

People live longer today than at any time in the past, and extended life expectancies should be accompanied by a high quality of life, especially for the elderly. However, after a certain age, the body is not able to promote natural repair after injury, even when a healthy lifestyle is adopted [1]. In addition to services required by older people, young people today are making greater use of surgical procedures for injuries caused by sports or accidents. Hence, an increase in the number of operations is leading to an increase in the market for orthopedic implants [2]. This is further amplified by the use of bone grafts in other medical fields, such as oncology, spine surgeries, and traumatology [3,4]. Severe trauma, osteoarthritis, and rheumatologic diseases damage human tissues beyond repair, and tumor resections—which are a common procedure in older people—can create irreparable damage or structural defects that do not heal. All of these events will inevitably add to the already enormous economic challenge facing healthcare and social systems [1,5].

Permanently implanted biomaterials are used to allow functional restoration and stimulate tissue regeneration and healing, mainly in total joint arthroplasties, such as total knee (TKA) and total hip arthroplasty (THA) [1,5,6]. Moreover, other enduring, provisional, or biodegradable devices are used to treat broken bones, lower back pain, osteoporosis, scoliosis, and other musculoskeletal diseases, further increasing the number of procedures and costs [5,7,8]. Advanced degenerative joint disease is treated by primary joint replacement; however, this means that surgical procedures and the incidence of related infections will increase as well, namely prosthetic joint infection (PJI) [6]. Thus, an interdisciplinary approach is needed in order to manage the complex issue of patients with PJIs [9,10,11,12]. PJI rates differ among studies, but they occur in about 0.5–1% of hip and shoulder replacements and about 0.5–2% of knee replacements [13]. PJIs can be treated with revision surgeries, but these are more prone to infection than primary surgeries, and are responsible for resource and cost increases [6]. These data are confirmed by national and international archives that considered thousands of patients surveyed over many decades [1]. Recently, it was estimated that the incidence of PJI in primary arthroplasties is 1–2%, and in revision arthroplasties, 4% [13]; these results are well in line with Italian results [14].

Underlying the pathogenesis of PJIs is the fact that prostheses are foreign bodies, and they can permit bacteria to attack the implant surface/bulk and necrotic tissues through the expression of adhesion proteins (Figure 1). When the bacteria produce extracellular compounds and biofilm is produced, the interaction between the bacteria and implant becomes irreversible [5,13]. Biofilm is a thin film of microorganism-embedded glycocalyx (exopolysaccharides), which coats the prosthesis surface and creates cavities that are convenient for nutrient passage [13,15]. It is a multicellular community that forms an appropriate habitat for bacteria. Several stages are involved in biofilm formation: (1) bacterial cell attachment to a surface, (2) cell proliferation, (3) biofilm maturation, and (4) detachment and dispersal. It should be noted that common pathogens and commensals, which may develop virulence, can be immersed in biofilm and use it as a substrate to grow [16,17]. In this context, bacteria are protected from antimicrobial drug action and they are metabolically less active, thus making any treatment difficult [1,5]. For these reasons, biofilm-associated microorganisms are problematic to diagnose and treat, and their detection from retrieved prostheses is also hampered [18].

Gram-positive bacteria are generally considered the most common PJI pathogens, accounting for approximately 65–81% of infections, and staphylococci are the most frequently isolated microorganisms in PIJs [10,19,20,21,22,23,24]. In particular, coagulase-negative staphylococci such as *Staphylococcus epidermidis*, *S. lugdunensis*, *S. capitis*, *S. hominis*, and *S. caprae* represent 30–43%, followed by *S. aureus* (12–23%), with methicillin-resistant *S. aureus* (MRSA) representing approximatively 15%. Other Gram-positive bacteria (*Enterococcus* spp., *Streptococcus* spp., *Clostridium difficile*, and diphtheroids) account for 11% [10,19,20,21,22,24,25]. Gram-negative bacteria (*Pseudomonas aeruginosa*, Enterobacteriaceae, *Proteus* spp.) are isolated in about 8% of cases. Among anaerobic bacteria, *Cutibacterium acnes* is the most frequent, while 10–12% of infections are caused by more than one microorganism [5,10,18,26].

Although implant configurations and applications may vary extensively, all of them attract microorganisms, representing niches for infection in vivo; this continuous microbial presence inhibits implant function and adds risk to human use. In particular, permanent internal implants face two challenges associated with their extended use in vivo: biomaterial-associated infection (BAI) and the lack of native tissue regeneration [1,17]. Biomaterials which are able to selectively allow human cell proliferation and simultaneously avoid microbial adhesion and subsequent biofilm formation are elusive; in fact, their composition usually permits both host cell and bacterial adhesion and growth, as the same adhesive mechanisms are used. Otherwise, structures and functionalization designed to prevent bacterial colonization do not effectively integrate with host cells and tissues [1,5,17]. A shift in strategy toward designing multifunctional biomaterials and their downstream application into clinical procedures is a key topic because biomaterials that are able to select host cells over microorganisms are urgently needed [1]. Because PJI is a major translational issue in healthcare, it should be managed by a multidisciplinary group that includes microbiologists, chemists, engineers, and orthopedic surgeons.

This review, carried out by the presentation of our research data and the examination of the relevant literature published from 2010 to date, is intended to provide an overview of the evolution of biomaterials in orthopedic applications while considering the properties and modification/functionalization leading to improvement in the prevention of microbial adhesion and biofilm formation, and the promotion host cell proliferation and colonization. In particular, we report the current state of our recent research compared with the latest literature data regarding modified biomaterials for replacement, such as metallic materials (titanium and titanium alloys) and polymers (ultra-high molecular weight polyethylene), and biomaterials for regeneration, such as poly(ε-caprolactone) and calcium phosphates as composites.

This current overview may offer a clearer insight into how biomaterial research and progress paves the way for the design and development of innovative devices for enhanced solutions to orthopedic clinical problems, mainly the effort to fight PJIs.

## 2. Biomaterials in Orthopedics

A biomaterial, alone or in a multifaceted application, is engineered to guide the connections of living system components to facilitate the course of any therapeutic or diagnostic technique [23]. Orthopedics is the most advanced field in the development and use of biomaterials, featuring their use to cure musculoskeletal disease and trauma. Orthopedic biomaterials are inserted into the human body in order to accomplish different biological tasks that is, to substitute or repair tissues such as bone, cartilage, ligaments, and tendons, and to direct bone repair. Thus, it is very important that the chosen material combines its features with the requirements of the application, considering the corrosive environment of the human body and the strict biological characteristics that the biomaterial needs to possess, in addition to the common ones (mechanical, chemical, and physical) [16]. Therefore, ideas of foreign body reactions (mainly due to wear debris), stress protection, biocompatibility, and—even more importantly—bioactivity and osteoinduction have progressively become requirements for biomaterials in implantable devices [8].

Over the last 60 years, biomaterial development and clinical obtainability for bioinert, bioactive and biodegradable materials, and materials designed to stimulate specific cellular responses at the molecular level, have progressed in successive phases. These stages should be interpreted not as being chronological but as being related to the design, as each one represents progress on the requirements and characteristics of the materials involved [8].

Fracture treatment has been transformed by the use of biocompatible alloys, including stainless steel, that were engineered in the last century to fix bone, mainly in the form of plates, screws, and pins. While bone damage was previously treated by suspending the limb in traction for a long time, today it is treated internally. Because long-lasting implants are necessary, more sophisticated biocompatible metal alloys are now available and used as first-choice materials for artificial joints [8,16,25,27,28,29,30,31]. Metals were followed by chemically manufactured polymeric materials, such as ultra-high-molecular-weight polyethylene (UHMWPE) or polymethylmethacrylate , which have been used progressively more often as bearing and connecting materials for artificial joints [8,32,33,34,35,36,37,38,39,40]. Thereafter, in order to replenish bone defects and to act as osteoconductive constituents, bioactive ceramics—primarily calcium phosphates (CaPs) or calcium sulfates—were introduced. Additionally, biocompatible and bioabsorbable polymers such as polycaprolactone (PCL), polyglycolic acid (PGA), polylactic acid (PLA), and polydioxanone (PDO) have been used as suture materials for a decade [3,41,42,43,44,45,46,47,48]. They are currently shaped as screws, pins, and plates, and absorbable polymers are used as scaffolds for cartilage and bone engineering [27].

In Table 1, a summary of various materials used in orthopedic applications is reported, together with the objects of our ten-year research project.

It is important to note that many factors influence the osseous response besides the implant’s chemical composition, including the biomechanical characteristics of the device. In fact, structural factors such as size, texture, macro- and microstructure, and material form are crucial to the cellular response [31]. Therefore, the clinical need has shifted to multifunctional biomaterials which are able to give specific responses simultaneously to colonization by different cells (osteoblasts, fibroblasts, macrophages) and microorganisms (i.e., bacteria, yeast, molds, and viruses). In order to accomplish these purposes, the current research aims to design simultaneous bioactive and antimicrobial biomaterials [28]. Thus, we can recognize two types of immunomodulating biomaterials: those for replacement, which are able to remain permanently inside and integrate with the body, determining slight inflammation and fibrous tissue development, and those for regeneration, which are able to guide/stimulate the initial new tissue formation and, thereafter, degrade in a precise way over time.

### 2.1. Biomaterials for Replacements

When artificial materials were first applied for biomedical applications, the only requirement was to “achieve a suitable combination of physical properties to match those of the replaced tissue with a minimal toxic response of the host” [8]. They were ‘inert’ so as to reduce the immune response and the foreign body reaction. Biomaterials for replacement are typically long-term (>20 years) or permanently implantable devices made of metals, polymers, or ceramics, very stable mechanically, and able to allow only a slight host reaction upon implantation [8].

Biologically inactive implants, typically characterized by native protein adsorption, can permit temporary matrix production and bind the biomaterial and the host [27].

#### 2.1.1. Metallic Materials: Titanium and Titanium Alloys

The first metallic materials used efficaciously during the twentieth century in orthopedic applications were stainless steel and cobalt–chrome-based alloys. Titanium (Ti) and its alloys became materials of interest in the biomedical field due to their outstanding characteristics [8].

Implanted biomaterials based on Ti and Ti alloys have been utilized extensively during orthopedic surgical procedures in the last 60 years, thanks to their biomechanical and biocompatible properties, mechanical properties, fatigue–corrosion resistance, low density, relatively low modulus, and resistance to corrosion by aggressive physiological fluids [4,28,31]. Their ability to resist corrosive environments is explained by the deposition of an adhesive TiO_2_ oxide layer at their surface [8].

Osteointegration is a very important aspect of implant anchorage to the surrounding bone, and great efforts are being made to design and optimize biomaterials’ surfaces to this end [8]. Ti implants demonstrate higher osteointegration when the surface is changed to encourage osteoblast migration and attachment [27]. Ti and its alloys can become tightly integrated into the bone, and this feature significantly improves the long-term performance of the implanted devices, reducing the risks of loosening and failure.

Recently, researchers pointed out that osseointegration is closely related to the surface properties of the biomaterials (i.e., roughness, wettability, and electrostatic charges) and their contacts with proteins/cells [8]. Orthopedic implants are osseointegrated through different steps that start with contact with the body fluids, followed by acute inflammation, cell-surface attachment, and—finally—new bone creation and remodeling at the implant surface [16,28]. The main challenges of Ti surfaces facing soft tissues are the development of well-anchored and -oriented soft tissue [29].

In contrast to the above-described properties, Ti has a low hardness, reducing its wear resistance and limiting its use in articulated joint parts. Other surface characteristics, for example, wear, are reduced due to the low shear resistance of Ti and Ti alloys, and these materials are not easy to process, whether through machining, forging, or heat treatment [8]. Additionally, the relevant issue of most Ti surface designs is to permit both host cell adhesion and proliferation, and microbial attachment [31].

In fact, Ti is well tolerated by the body as long as the implant is in bulk form, mechanically stable, and non-infected. If the latter circumstances are not encountered, the implants can be accompanied by an acute/chronic inflammatory reaction, osteolysis, loosening, and failure. Stainless steel, cobalt, and Ti alloys can be corroded by osteoclasts, causing metal ion release and thus creating an inflammatory environment; the degree and clinical importance of this reaction is yet to be fully explained [28,31]. The high infection frequency associated with metallic biomedical implants and devices has a relevant effect on human health and healthcare costs [65,66,67]. Medical-device-related infections can be reduced by inhibiting microbial adhesion on the surfaces of these medical devices. This goal might be achieved with the modification of the surface features of the implant, using surface coating techniques that combine biocompatible and mechanical characteristics, making the surface less attractive for bacteria [30].

The combination of osteogenesis to encourage osseointegration and antimicrobial action to discourage the microbial presence on Ti surfaces can be accomplished by introducing inorganic bioactive layers made of a suitable inorganic agent, such as silver (Ag), copper (Cu), zinc (Zn), or cerium (Ce), provoking an antimicrobial effect via the contact-killing or release-killing modes, or both [28,49,68]. At the same time, bioactive performance can be achieved on Ti and Ti alloys with another type of coating made of foreign material (apatite or bioactive glasses), using electrochemical procedures or chemical surface treatments [28].

#### 2.1.2. Polymers: Ultra-High-Molecular-Weight Polyethylene

UHMWPE is a particular type of polyethylene (PE) with a remarkably high molecular mass. UHMWPE, similarly to most polyethylenes, is a semicrystalline polymer composed of at least two interpenetrating phases: a crystalline phase in which the macromolecules are well-organized crystalline lamellae, and an amorphous, disordered phase, possibly intercalated by a partially ordered, so-called ‘all-trans’ interphase. The polymer microstructure, in addition to the molecular mass, is a key factor in the definition of its various features (i.e., physical, chemical, and mechanical), which together represent a crucial combination of mechanical properties and high wear and abrasion resistance [32,33,35,39]. Additionally, UHMWPE is characterized by chemical inertness, lubricity, low friction, high impact strength, excellent toughness, low density, ease of fabrication, biocompatibility, and biostability [8,32,38].

Since its initial application in 1962, UHMWPE has been used in orthopedic surgery as a bearing material in the liners of acetabular cups in total hip arthroplasties, in the tibial insert and patellar element in TKAs, and as an insertion in intervertebral artificial disc replacement. In fact, UHMWPE—due to its unique features—is applied as a soft insert combined with stiffer materials (mainly ceramics and metals) in hip joint substitution, showing very good tribological performance in terms of friction and wear [33]. It is considered the gold standard for orthopedic implants, and its future demand will increase by nearly 171% by 2030. Unfortunately, the components of this material frequently have a reduced lifetime; wear and in vivo damage are the most limiting factors [36]. Recently, the growing demand for joint arthroplasty in young and active patients has led to the progressive development of bearing materials with increased performance and durability [32]. Indeed, aseptic loosening happens when a component of an implant made of polyethylene—due to wearing—leads to debris, thus making a revision procedure necessary. Two main complications are associated with UHMWPE: particulate wear and the resulting osteolysis due to the wear debris of the implant, and delamination wear due to oxidation caused by sterilization techniques that use high-energy irradiation followed by storage and packaging in the presence of air (oxygen) [8,37]. These issues have been resolved, and no longer represent clinical problems. In 1998, the market for polyethylenes for surgical procedures was improved with highly cross-linked and thermally treated polyethylene (HXLPE), which was characterized by high wear resistance. First-generation HXLPE was able to reduce wear debris, and thus the osteolysis caused by the inflammatory response to worn elements. It was designed with the concept of manipulating the cross-linking process and, at the same time, reducing the oxidative process caused by the free radicals produced during polyethylene irradiation [32,35]. In order to reduce or remove the remaining free radicals, the highly cross-linked, wear-resistant UHMWPE used for clinical purposes today is made using irradiation followed by thermal treatment. Conversely, melting diminishes the irradiated polymer’s crystallinity and mechanical features.

Current research is aimed at delaying the occurrence of oxidation by adding appropriate stabilizing compounds that can slow the oxidation processes through interference with radical species’ reactivity without affecting the chemical, physical, and mechanical properties, thus prolonging the material’s lifetime [33,34,37]. Vitamin E (VE, alpha-tocopherol) appears to be the perfect candidate because it is already present in the human body as a natural antioxidant in physiological processes [34,39]. Hence, the irradiation of vitamin-E-stabilized PE led to a cross-linked UHMWPE, namely second-generation HXLPE stabilized with vitamin E, which was distributed in the market in 2007–2008, and had enough cross-link density to guarantee high wear performance and long-term oxidative stability which was able to withstand extended wear and preserve mechanical features [32]. Additionally, vitamin-E-blended UHMWPE may have the advantage of a decreased inflammatory response to its wear particles [34,38,40]. In addition to wear and component degradation, infection is still a relevant obstacle to total joint replacement. In fact, many different aspects can contribute to microorganism attachment to an implanted device, such as microbial virulence, nutrient obtainability, the host immune system, and the chemical surface characteristics of the device itself. Thus, the vitamin E functionalization of UHMWPE might increase oxidative resistance and influence these bearing surface features, and therefore plays a role in the susceptibility of the inserted UHMWPE to infections [40].

### 2.2. Biomaterials for Regeneration

Tissue engineering is a new development that opens numerous research possibilities in the area of regenerative medicine, an interdisciplinary field that involves the complementary skills of engineers, chemists, physicists, biologists, and physicians [8,48]. Biomaterials for regeneration are designed to re-establish the lost structure and function of damaged tissue. These biomaterials should degrade in a period of several days to months while promoting host tissue regeneration that includes the regenerated matrices; they combine several important features, namely bioactivity, biodegradability, and bioresorbability [27]. In addition, these characteristics might be combined with the ability to signal and stimulate specific cellular activities and behaviors [8,48].

#### 2.2.1. Poly(ε-caprolactone)

The main purpose of biomaterials in tissue regeneration is to allow the process of new synthesis at the site of a bone loss, and to be “resorbed and replaced over time” with newly generated bone tissue. Thus, they need to possess the following features: (i) osteoinduction, promoting the change from progenitor cells into mature osteoblasts; (ii) osteoconduction, supporting/facilitating new bone tissue growth; and (iii) osteointegration, helping to integrate the new bone with the nearby bone tissue. Additionally, the bone grafting biomaterial, whether it is made with biological or synthetic materials, should be chemically and mechanically stable in the host environment, non-thrombogenic, and easily sterilizable, and should possess adequate manufacturability [41,42]. 

Deficient tissue can be restored and substituted by a three-dimensional porous construct called a ‘scaffold’ that—in the short term—allows cell attachment, and thereafter promotes cell spread and differentiation [44,48,69]. Based on the processing techniques used, these scaffolds may possess highly modifiable mechanical properties, porosity, and degradation degrees [45], and they should meet specific requirements for in vivo applications [47,48]. Because new bone generation is a slow event, a slower degradation is necessary to guarantee the scaffold’s biomechanical stability preceding tissue remodeling [47,48]. A scaffold designed for bone regeneration requires a structure characterized by interconnected pores in order to facilitate nutrient and metabolite passage, to allow vascularization and the production of new bone, and to create an advantageous microenvironment for cell adhesion, proliferation, and differentiation without provoking any adverse effects. In addition, the scaffold must be degraded simultaneously with the repair/regeneration of new bone. A scaffold should exhibit similar mechanical properties to the natural tissues in order to offer the proper rigidity and stiffness [42,44,69]. The micro-and macrostructure of the construct varies with the processing procedure; pore distribution, interconnectivity, and size are of great importance. One of the main goals behind the different strategies involved in research into functionalized bioabsorbable polymeric, CaPs, or composite substrates as scaffolds is the ability to stimulate cell activity while discouraging microbial colonization [8].

Polymers for bone implants are characterized by their biocompatibility, modeling flexibility, low weight, and ductility. Biodegradable polymers of natural (chitosan, poly(2-hydroxyethyl- methacrylate), hyaluronic acid, and other hydrogels) and synthetic (polyglycolide, PLA, PDO, PCL, polyhydroxybutyrate, polyorthoester) origin have been extensively studied [3,8,41,42,45,57,60,61]. Natural polymers have the benefits of biocompatibility and degradation, whereas synthetic ones offer suitable prospects as enhanced materials to modify physicochemical and topographical properties in order to increase cell proliferation and new tissue growth [42]. Natural polymers are challenging to re-engineer; no modification of the polymer structure is permitted. Thus, synthetic polymers have attracted interest for biomedical applications, mainly bone tissue engineering, thanks to their biodegradability and biocompatibility [3,70]. Many medical implants made with these synthetic polymers have been approved for clinical use in humans [71]. However, the products of the degradation of synthetic-polymer-based biomaterials also present low rigidity, and at times may pose certain issues, such as the buildup of acid in the surrounding tissue [42].

PCL, an aliphatic, semicrystalline, and synthetic polymer, is the most widely diffused biodegradable and noncytotoxic polymer; it is used in many Food and Drug Administration -approved surgical implants to assist in bone regeneration, and in drug delivery devices for regenerative medicine due to its interesting mechanical properties and degradation kinetics [3,46,47,48]. It displays high toughness and mechanical strength, and is relatively elastic, along with adequate biocompatibility; it is extremely compatible with osteoblasts [41,42]. The disadvantages of PCL in bone tissue engineering are its slow degradation rate (2–4 years in living tissue)—which occasionally negatively influences the bone tissue regeneration process [42]—and its poor hydrophilicity [45]. PCL also has reduced mechanical stability [44].

#### 2.2.2. Calcium Phosphates and Composites

The use of CaPs, known as bioceramics, started in the 1970s with their application as bone defect fillers due to their structural strength, compression resistance, high biocompatibility, and tissue response. The similarities between the bone mineral phase and their structural/surface properties are responsible for their good bioactive features, which bind bone with no fibrous connective tissue interface deposition. Bioactive ceramics are also biocompatible and osteoconductive [8]. CaPs can offer structural integrity to the implant in order to keep it in place and intact until the new bone grows; they are also soluble, such that they can be reabsorbed by the body, allowing the new bone to replace the construct [24,41]. The disadvantages of CaPs are related to their poor mechanical properties, namely their fragility and low stiffness [72].

Hydroxyapatite (HA), β-tricalcium phosphate (β-TCP), their derivatives, and their combinations are the most commonly used ceramics. Their physicochemical properties are tunable with different synthesis methods.

HA is the major crystalline form of CaP, representing about 70% of bone tissue and determining its mechanical strength. HA has been demonstrated to possess superior bioactive features, and it is chemically stable because it has low solubility rates under in vivo conditions with respect to TCPs; hence, following HA implantation, it remains integrated within the newly formed bone. In contrast, TCPs such as β-TCP are fully reabsorbed under physiological conditions [1,6,7]. The latter is extremely biocompatible, and produces a resorbable interlocking network within the defect site in order to promote healing. CaP compounds have gained attention in biomedical applications, and are mainly used as bone substitutes due to their desirable noncytotoxic, noninflammatory, and nonimmunogenic properties [46].

Therefore, biphasic calcium phosphate (BCP) compounds containing HA and β-TCP with different HA/β-TCP ratios have been extensively developed in order to enhance both biomaterial properties and resorption rates [43].

Composites—combinations of two or more materials—might have variable compositions/features, and can be adapted to improve mechanical properties or bioactivity [45]. The new class of biomaterials known as composite scaffolds permits the engineering of biomaterials with desired mechanical and physiological characteristics by controlling the type, size, fraction and morphology, and reinforcing phase organization [44]. The composites should also allow host cell activities within the biomaterial, which are important for tissue regeneration compatibility [42]. The development of hybrid or composite biomaterials brings together the advantages of both materials, producing biomaterials with properties which are superior to those of the raw material used. A good strategy to improve the performance of polymers such as PCL could be the incorporation of another material as a filler, for example BCP [41].

## 3. Antibacterial and Anti-Adhesive Compounds

A key problem for public health worldwide is the increasing antimicrobial drug resistance of pathogenic bacteria (e.g., MRSA), with the consequence of increasing surgeries, complications, and hospital stays due to treatment failure and poor clinical efficacy. This situation demands urgent action against the threat of drug-resistant bacteria, avoiding the emergence of drug resistance and controlling the spread of drug resistance [50,73]. The European strategy is to employ biomedical research to develop alternative solutions to inhibit bacterial infections and biofilm formation in order to reduce antibiotic resistance and the potential hypersensitivity, allergy, and toxicity associated with it [16,28].

Thus, in order to avoid the direct use of antibiotics, an alternative approach should be the development of biofunctional fillers which can simultaneously promote bacterial killing and bone regeneration in order to enhance the therapeutic efficacy of bone tissue engineering [73].

### 3.1. Metal Ions

The advantages of inorganic antibacterial agents (introduced into bioactive thin or thick coatings, as well as in modified surface layers, or present in the bulk of the structure) are the broad activity spectrum (optimal for both the treatment of polymicrobial infections and the prevention of contamination from unknown pathogens) and low resistance development. On the other hand, the application of this innovative approach may be complex, with some complications related to certification, regulation, and finding the proper therapeutic window [28]. In fact, dose-dependent antibacterial activity is often reported for inorganic antibacterial agents such as Ag, Cu, Zn, and Ce as metallic ions or nanoparticles [30,74,75,76].

Silver, known from ancient times to have antimicrobial properties but neglected after the discovery of penicillin, has recently been considered once again [77]. It is widely used in the pharmaceutical industry and in many consumer products thanks to the broad and strong spectrum of its activities against both Gram-positive and Gram-negative bacteria and fungi, and the lack of associated resistance [16,52,73].

Silver causes the death of microorganisms due to the presence of silver ions that alter the function and structure of proteins in the bacterial cell wall, leading to their rupture. Silver ions bind to and alter several enzymes which are crucial for cellular respiration and metabolism, and interfere with DNA through cell division and replication [31,50,75,78,79,80]. Furthermore, Ag can release Ag+ ions from the biomaterial, exerting its antimicrobial effect via the contact-killing mode and/or release-killing mode [28]. Silver shows a dose-dependent antimicrobial effect and a slight difference in the susceptibility of Gram-negative and Gram-positive bacteria. Gram-positive cell walls are about ten times thicker, with multiple murein layers and teichoic acid, and protect the cell against silver ions [68,78].

Other metal ions (e.g., Cu, Zn) possess an analogous mechanism of action [50]. It is known that silver is more toxic towards prokaryotes than towards mammalian cells, which implies a therapeutic option where mammalian tissue is not harmed but where bacteria are killed [78]. The bacteriostatic or bactericidal effect of silver or copper metals, or their ions—used as admixtures of carbon coatings—is primarily related to the effect exerted on the microbial cell and not to the change of surface morphology [78,81]. Zinc oxide (ZnO) nanoparticles possess antibacterial activity against a broad spectrum of Gram-positive and Gram-negative bacteria because they generate reactive oxygen species (ROS), particularly H_2_O_2_, causing oxidative stress accompanied by nanoparticle accumulation in the cell matrix of bacterial membranes [82].

Conflicting results concerning the cytotoxicity of metal ions such as silver ions have been reported in the literature [16,28]. It is noteworthy that metal ions offer protection from microbial adhesion and biofilm formation but can also affect the surrounding tissues, including positive effects such as the stimulation of osteoblasts, leading to improved osseointegration. On the other hand, they may also exert cytotoxic effects. Ag^+^ is also associated with mitochondrial dysfunction, disturbed membrane integrity, ROS induction, and DNA lesions [31,52,75,78,83]. The release rate and loading amount of Ag should be controlled accurately in order to avoid transient cytotoxicity when used at high concentrations that may exceed the biocompatible level and cause long-term toxicity due to accumulation in the kidney [73].

In preclinical animal trials, the Ag^+^ released from orthopedic implants accumulated in organs (i.e., liver, kidneys, spleen, brain), and serum concentrations increased significantly after the implantation of silver-coated endoprostheses. It may also be assumed that the implantation of silver-coated endoprostheses resulted in the accumulation of ionic silver in the peri-implant tissue, impairing the viability and differentiation of osteoblasts and osteoclasts. Therefore, despite negative findings in conventional clinical laboratory tests and standard histological evaluations, locally or systemically elevated silver levels may compromise cell function and pose a potential risk for human organ systems [84].

### 3.2. Essential Oils

The interest in medication with essential oils (EOs) is due to the belief that they represent safer and more dependable natural components compared to synthetic drugs, which can sometimes have adverse effects [85,86].

EOs are secondary metabolites formed from tens to hundreds of molecules belonging to the class of terpenoids and phenylpropane derivatives, or complex mixtures of terpenoids containing sesquiterpene and monoterpene, and their oxygenated derivatives. EOs may also incorporate a variety of other molecules, such as fatty acids, oxides, and sulfur derivatives. Both the terpenoid and phenylpropanoid families, identified as the principal constituents of several EOs, can constitute 85% of the total concentration of the oil [87]. Some of the most representative plant families containing EOs are *Lamiaceae*, *Apiaceae*, *Myrtaceae*, *Zingiberaceae*, *Lauraceae*, *Rutaceae*, *Asteraceae*, and *Cupressaceae* [88].

EOs such as peppermint, cinnamon, lemon, and clove—due to their anti-inflammatory and antioxidative effects, and their capacity to inhibit bacteria (Gram-negative and Gram-positive), fungi [85], and viruses [89]—have been applied to accelerate the wound healing process.

Little is known about the mechanisms of EO antimicrobial activity; however, the literature proposes pathogen membrane permeability alteration, cytoplasm, enzyme and protein chemical modifications, and changes in microbial cell conformation [90]. Furthermore, the increase in the plasma membrane’s permeability to cellular metabolites appears to be due to the presence of terpenes, alcohols, aldehydes, and esters [88], the presence of which explains the absence of microbial resistance or adaptation to their pharmacological properties [91].

The applications of EOs in medicine are limited as a consequence of their volatility, low stability, and high sensitivity to environmental factors; however, in the past decade, research has focused on studying the therapeutic properties of biomaterials containing EOs [85]. In fact, EOs are generally recognized as safe, and can act synergistically with other compounds—promising factors for their use as bioactive compounds [87,92].

*Mentha piperita*, known as peppermint, has a long history as a therapeutic remedy [88]. It has been widely used in folk remedies and the food, cosmetics, and pharmaceutical industries. Peppermint (P)-EO is used for coughs, colds, mouth sinuses, pain relief, headaches, skin and digestive affections, and as a muscle relaxant [86]. This plant also shows chemopreventive, antioxidant, and antimutagenic potentials, renal activity, and anti-allergenic effects [93]. The gas chromatography–mass spectrometry analysis of peppermint EO showed the presence of menthol, menthone, limonene, isomenthone, menthylacetate, carvone, pinene, 1,8-cineole, pulegone, piperitone oxide, and micene [85,87]. Menthol and menthone are constituents which are responsible for the elimination of bacterial pathogens—both Gram-positive and Gram-negative bacteria. They act on the cell membrane, causing substantial morphological damage and destabilizing the microbial membrane [86]. In particular, they disrupt the lipid fraction of microorganisms’ plasma membranes, resulting in alterations in membrane permeability and the leakage of intracellular materials. The oil might also penetrate the cell and interact with intracellular sites which are critical for antibacterial activity [94]. Furthermore, P-EO exerts effects against fungi (*Candida albicans*, *Aspergillus albus*, and dermatophytes) and viruses (*Herpes simplex virus* 1 and 2) [85,86,87,88,91,92,93].

### 3.3. Vitamin E

Vitamin E is a natural product which is found in a wide range of plants and microorganisms, and is an essential source of structural diversity and complexity for the development of novel antimicrobial agents, especially those designed to fight resistant bacteria. Its derivatives also possess a broad spectrum of biological activities such as antioxidant, antiproliferative, cholesterol-lowering, and neuroprotective effects [95], and it is known to decrease ROS and TNF-α expression and prevent cell apoptosis [96].

Recently, a work confirmed vitamin E’s action in inhibiting *P. aeruginosa* biofilm formation [97], in agreement with Vergalito et al., who demonstrated that it could interfere with and reduce biofilm formation in a wide range of human pathogens, including *P. aeruginosa, P. putida*, *E. coli*, *K. pneumoniae*, *A. baumannii*, *Proteus mirabilis*, *S. epidermidis*, and *S. aureus* [98]. Previously, Chen and colleagues, investigating the bactericidal activities of macrophages exposed to particles with and without vitamin E, demonstrated an enhanced antistaphylococcal activity of cells in the presence of the vitamin compared to controls [96].

## 4. Biomaterials Coupled with Anti-Adhesive and Antimicrobial Agents

In this section, different combinations of biomaterials and antimicrobial compounds are presented. We refer to previous reviews and excellent research papers, but this analysis is not intended to be exhaustive. A summary of the antimicrobial/anti-adhesive properties of biomaterials used in orthopedic applications—modified with different compounds assayed against different bacterial species—is provided in Table 2. In Table 3, the cytotoxicity results for modified biomaterials are described. In both Table 2 and Table 3, the objects of our decennial research are reported. Figure 2 schematically presents the positive effects following the implantation of a prosthesis that was modified in order to improve its characteristics. The development of biomaterials that incorporate antimicrobial agents could reduce microbial adhesion (via contact-killing or release-killing modes, or both) and biofilm formation on these surfaces, thereby preventing medical-device-related infection (Figure 2).

### 4.1. Biomaterials for Replacements Coupled with Anti-Adhesive and Antimicrobial Agents

One strategy to prevent the development of PJIs is the surface modification of titanium substrates. In fact, the grafting of an antimicrobial agent directly onto an implant surface can prevent bacterial colonization and subsequent infection. On the other hand, treatment can be administered via the localized delivery of a therapeutic agent, improving its efficacy in the case of infection caused by biofilm development on the implant’s surface. Moreover, a reduction in the amount of the antibacterial agent used, mainly with regard to inorganic antibacterial compounds (such as Ag, Cu, Zn, or EOs), and its localized delivery (only where needed) can reduce the side effects of systemic application [31,50,74].

Regarding titanium, in prior research [99] we evaluated the effect of a modified titanium alloy (Ti6Al4V) surface on the induction of bone tissue integration and the prevention of infections at the implant site and around it. In this study, we used silver nanoparticles due to their broad-spectrum antibacterial activity and low resistance development [112,113,114]. Initially, we produced titanium surfaces with silver nanoparticles embedded in the oxide layer of the titanium surface, which were able to release silver ions, inducing antibacterial behavior without the use of silver micro-aggregates that could generate cytotoxic effects [115]. Thereafter, several surface features of the silver-treated samples were verified: the nanotextured oxide layer with the functionalities was suitable for osseointegration, and the high-density hydroxyl groups were able to induce surface bioactivity and high wettability, as well as to allow further surface modifications. The surface oxide layer was tightly adhered in order to guarantee its stability during implantation [50,115,116,117,118]. In order to control the reduction reaction of the silver ions and the size of the silver nanoparticles, we tested different additives: glucose (GLU) was selected for its reducing activity; polyvinyl alcohol (PVA) was selected as a nanoparticle stabilizer; and gallic acid (GA), starch (ST), and tannic acid (TA) were selected as reducing agents and nanoparticle stabilizers [119,120,121].

The second step was to assess the antibacterial behavior of the different titanium surfaces: chemically treated (CT) Ti6Al4V and Ti6Al4V mirror-polished samples were used as controls, whereas Ti6Al4V CT-Ag, Ti6Al4V CT-TA+Ag, Ti6Al4V CT-GA+Ag, and Ti6Al4V CT-GA+PVA+Ag were assayed as treated samples [99]. We observed a significant inhibition halo against *S. aureus* for all of the treated samples containing silver on the surface. The bacterial adhesion test evidenced a moderate 33%, although not statistically significant, reduction in staphylococcal adhesion on the CT surface, without the introduction of silver ions, with respect to the polished surface. This result underlined, for the first time, the ability of a Ti6Al4V surface nanotexture to limit bacterial adhesion, compared to a mirror-polished control, even without the addition of silver. Additionally, significant (*p* < 0.05) antistaphylococcal activity on the silver surfaces, with a maximum bacterial adhesion reduction of 97% for Ti6Al4V CT-Ag, was obtained. A statistically significant (*p* < 0.05) decrease in colony-forming-units (CFUs) was also achieved in the cultural broth, demonstrating the release of silver from the modified titanium surfaces, independent of the type of additive employed.

In conclusion, we deduced that the surfaces studied in our work [99] possessed both suitability for osseointegration due to the nanotextured oxide layer and antibacterial properties due to the silver nanoparticles embedded in the surface oxide layer and silver ion release.

The above-reported results [99] are in line with those of Cochis et al. (2020) [16], in which doped bioactive materials (Ti6Al4V alloy and a silica-based bioactive glass with added strontium and/or silver ions) reduced the surface colonization of drug-resistant *S. aureus*, and were cytocompatible with the bone progenitor cells used to represent the self-healing process [16]. Similar antibacterial data were also achieved by Li et al. (2016) [122] with silver and by Zong et al. (2016) [123] with copper, as reported in a recent review [28]. It is noteworthy that Zong et al. (2016) [123] observed that Cu-doped TiO_2_ nanotubular surfaces exhibited good long-term antibacterial properties (for 1 month) against *S. aureus* due to the sustained release of Cu^2+^ ions. Cu doping also upregulated the secretion of vascular endothelial growth factors (VEGF) by endothelial cells compared to pure TiO_2_ nanotube arrays, which were used as a control. Hence, this multifunctional implant could accelerate tissue healing processes while simultaneously being able to kill bacteria and promote angiogenesis [28].

Titanium coated with diamond-like carbon (DLC) was studied by Zhao et al. (2018), who highlighted that it reduced both *E. coli* and *P. aeruginosa* attachment by up to 75% [30]. DLC coatings, with the addition of copper or silver, reduced the amount of *E. coli* that developed on their surface, and their use as a coating for orthopedic implants is associated with a decrease in the biological response of endothelial cells [75].

In another work [105], we studied *S. aureus* adhesion on a surface-modified CT Ti6Al4V alloy coupled with molecules from P-EO (CT_Mentha oil). For the first time, a homogeneous and continuous oil coating was created, containing a prevalence of oxygenated compounds (menthol, hydroxyl-menthofuran, and menthyl-acetate).

The reduction in *S. aureus* adhesion on the coated and functionalized CT_Mentha oil samples was significantly (*p* < 0.05) greater than that of the untreated controls; that is, CT and polished Ti6Al4V. On the contrary, no effect on planktonic bacteria was noted, confirming that no release of the P-EO by the coated/functionalized surfaces occurred in aqueous media, as was additionally demonstrated by the absence of an inhibition halo on agar. According to these data, the biofilm growth ability tests on the CT_Mentha oil samples confirmed that the oil was successful in reducing bacterial viability after 24–48–72 h in comparison with CT (*p* < 0.05) [105]. Larger amounts of grafted antibacterial compounds (phenols and terpenoids) on coated samples resulted in a significant reduction in both the adherent staphylococci and biofilm formation, thanks to their wide-ranging and diverse activities that target different mechanisms. For example, phenols interfere with the N-acylhomoserine lactone-mediated quorum-sensing signaling that is fundamental for the recruitment of floating bacteria [124,125]. Overall, these results highlighted that peppermint EO can be considered a promising coating for implant surfaces in order to prevent bacterial adhesion and biofilm [105].

Regarding CaPs, the antibacterial activity of hydroxyapatite coated with peppermint essential oil was tested by Badea et al. (2019) [86] against different Gram-positive bacteria (methicillin-resistant and sensible *S. aureus* and *Enterococcus faecium*), Gram-negative bacteria (*E. coli* and *P. aeruginosa*), and a fungal strain of *C. parapsilosis*. The results revealed that the antimicrobial activity of hydroxyapatite coated with peppermint essential oil increased significantly compared to that of hydroxyapatite alone [86]. In another work, the results regarding hydrophilic polyurethane-based wound dressings containing peppermint extract highlighted that the release of peppermint extract from nanofibers continued for 144 h, with an antibacterial efficiency of 99.9% against both *S. aureus* and *E. coli* [93].

Finally, in a recent study, the antimicrobial properties of novel bone cements modified with peppermint essential oil were assayed against *P. aeruginosa* and *S. aureus*. The samples loaded with 5% peppermint essential oil, as the active component, generated clear inhibition zones compared to the control samples for both strains, demonstrating peppermint oil’s inhibitory effect. These results were confirmed by the quantitative assessment of the adhesion and biofilm formation on the modified bone cements. Furthermore, the peppermint essential oil-enriched samples were not toxic to fibroblast-like cells and, conversely, stimulated cellular metabolism, with a significant increase in proliferation [85].

In a previous study regarding UHWMPE [107], we hypothesized that the adhesion of different *S. epidermidis* (biofilm-producing or nonproducing) strains could be different on polyethylenes (PEs) with modified chemical surface features. We assayed standard UHMWPE (PE), PE blended with 0.1% *w*/*w* vitamin E (VE-PE 0.1) or 0.5% *w*/*w* (VE-PE 0.5), and oxidized PE (OX-PE) for staphylococcal adhesion, leading to the observation that surface oxidation facilitates *S. epidermidis* adhesion.

In fact, a significantly higher (*p* < 0.01) incidence of bacterial adhesion was detected on OX-PE, whereas a significant decrease (*p* < 0.01) in the number of dislodged biofilm staphylococci was registered on VE-PE 0.1 in comparison with that observed on PE, within 24 and 48 h. We demonstrated that the effect of vitamin E was independent of its concentration, as no significant differences in adhesion were observed between VE-PE 0.1 and VE-PE 0.5 [106,107].

The outside of the bacterial cell plays a relevant role in its binding, as it directly interacts with the substratum surface; thus, in this context, we can state that the different chemical surface properties of PEs demonstrated an active role in their behavior [40]. It is widely known that surface hydrophilicity both influences protein binding ability and directs the interaction between adsorbed proteins and microorganisms. There is also a correlation between increased hydrophilicity and enhanced protein binding through hydrogen bonding interactions [4,126]. Numerous studies have proven that the initial bacterial adhesion to a biomaterial is controlled by nonspecific interactions, including hydrophobic interactions [107]. Hydrophobic biomaterial surfaces generally show great adhesion, but this phenomenon is strongly affected by adhered proteins. The presence of plasma proteins leads to a reduction in bacterial adhesion, but sometimes polar hydrophilic surfaces may result in increased bacterial adhesion and aggregation, which is in line with our observations [35]. PE is sensitive to oxidation [32], and PE oxidation increases surface hydrophilicity through the incorporation of polar, oxygen-containing, and functional groups. In these circumstances, polar hydrophilic surfaces may result in increased bacterial adhesion and aggregation, in agreement with our findings. In contrast, the antioxidant effect of vitamin E results in a lower surface hydrophilicity and, consequently, lower bacterial adhesion [35,40,110,127,128].

Furthermore, with regard to our studies on UHMWPE [108], we also evaluated the adhesion of other bacteria (i.e., *S. aureus* and *E. coli*, both ATCC and clinical strains) to PE, VE-PE, and OX-PE. Even in this case, the surface oxidation and the protein absorption facilitated microorganism adhesion, confirming what we previously reported with *S. epidermidis* [107]. In fact, significantly lower *S. aureus* and *E. coli* adherence was achieved on VE-PE compared with that on both PE and OX-PE after 48 h. These data underline a further benefit of VE-PE, which already possesses favorable mechanical, wear, and oxidation properties.

Despite the fact that the potential effects of VE on infection have been investigated, at present, it is still unclear how it affects bacterial adherence to PE [129]. Nevertheless, it is well known that when it is added to UHMWPE, it does not have any cytotoxic effects and acts as an effective anti-inflammatory agent [130]. These results were further confirmed by Chen et al. (2017) [96], who stated that VE-PE particles were able not only to induce less apoptosis in macrophages but also to trigger macrophage proliferation to form proper immunity within peri-implant tissue against *S. aureus*, showing high biocompatibility [96].

Finally, another of our studies [109] also confirmed a decreased adhesion to VE-PE for *C. albicans*. Furthermore, a similar reduced adhesion pattern was proven for the cross-linked PE, which was recently introduced into clinical practice, and was subjected to the cross-linking process in order to improve wear resistance and thermal treatments [106]. After 48 h of incubation, all of the tested microorganisms—*S. epidermidis*, *S. aureus*, *E. coli*, and *C. albicans* (both ATCC and clinical strains)—revealed lower adhesion on the cross-linked PE, with respect to PE alone. Various microbial species, and even strains within one species, can exhibit widely differing virulence, adhesion, and growth patterns on different biomaterial surfaces, as well as different antibiotic susceptibilities. Therefore, bacterial strains for in vitro testing must be carefully selected in order to prevent a bias toward favorable results [1]. This is the reason why, on UHWMPE, we assayed different Gram-positive and Gram-negative bacteria, and yeasts [107,108,109]. Bidossi et al. (2017) [110] demonstrated that α-tocopheryl phosphate and α-tocopheryl acetate were able to affect considerably *S. aureus* and *S. epidermidis* adhesion, and to prevent biofilm formation. Furthermore, they also showed an inhibitory effect on the development of staphylococcal biofilm on titanium discs [110,128].

Additionally, the VE stabilization of UHMWPE induced two synergic effects on osteoblasts. It improved osteoblasts’ capacity to respond to oxidative stress by inducing the cellular defense mechanism, and also—thanks to the antioxidant effect—influenced osteoimmunological factor secretion, thereby stimulating bone protection [111].

### 4.2. Biomaterials for Regeneration Coupled with Anti-Adhesive and Antimicrobial Agents

An initial study on composites made with PCL and CaPs was conducted by our research group [53]. Firstly, we designed and produced a bilayer scaffold made of a dense core portion, planning to include a load-bearing function and a porous surrounding in order to provide biological performance. Subsequently, we dealt with the hybridization of the robocasting technique through conventional methods for the processing of polymer parts, with the aim of offering increased scaffold functionality (i.e., antibacterial and antiadhesive features). The anti-*S. aureus* capacity was obtained through the addition of silver ions to the polymer layer. The Ag^+^-free samples (control) exhibited a significant number of adhered bacteria, indicating the poor antimicrobial behavior of either the ceramic core or the polymer layer. In contrast, the number of adhered staphylococci greatly decreased in the presence of Ag^+^-doped samples, evidencing the high effectiveness of the antibacterial layer in the reduction of cell adhesion to the total hybrid scaffold. This decrease was further demonstrated by field emission scanning electron microscopy (FESEM) micrographs. Once again, the Ag^+^-free control samples showed a high accumulation of planktonic bacteria grown in the broth after 24 h incubation, while a significant decrease in staphylococcal growth was noted for the Ag^+^-functionalized samples. In this study, we identified a hybrid material that could potentially be useful for tissue engineering applications, in the form of trilayer samples in which each layer had special functions: loadbearing for the dense core layer, osteoconductivity and osteoinductivity for the intermediate porous layer, and antibacterial and drug delivery for the outer composite layer [53].

In a recent research work [46], we studied novel PCL-based biomaterials for use as bone scaffolds, designed through modification with both BCP and silver nitrate in order to impart bioactive/bioresorbable and antibacterial properties, respectively. The results demonstrated silver-enriched PCL biomaterials with strong anti-adhesive and anti-staphylococcal activity, with a significant (*p* < 0.0001) reduction in *S. aureus* adherence onto Ag^+^-doped samples. These results were also supported by FESEM analysis. A comparably significant (*p* < 0.0001) reduction trend was also recorded for planktonic bacteria, with a decrease in the bacterial count in the presence of the Ag-doped PCL-based biomaterial. Therefore, the adhesion assay results, together with the inhibition halo data, validated the strategy of antibacterial Ag^+^ release from the Ag-enriched PCL scaffolds into the surrounding medium as an effective tool for the eradication of bacterial contamination from the surgery site, further confirming the antistaphylococcal action of Ag^+^ [46,131,132]. In this report, preliminary tests on sarcoma osteogenic-2 (SaOS-2) cells were described; PCL and BCP/PCL were biocompatible, but cell viability decreased in the presence of Ag-enriched biomaterials [46].

In the biomedical field, various approaches have been applied to provide antimicrobial properties to PCL [47,52,82,131,133,134]. Most of the literature data pertain to skin tissue engineering. In the research of Afghah et al. (2020) [100], 3D-printed PCL scaffolds were combined with silver particles and 1,3-propylene succinate , revealing the good antibacterial activity of these samples against *E. coli*, *P. aeruginosa*, *S. aureus*, and *C. albicans*. However, the Ag concentration did affect human dermal fibroblast viability unless its concentration was reduced to 2.5 or 1% [100]. In another study [101], electrospun PCL fibers were core-shelled with silver, and, once again, both antibacterial properties and noncytotoxic effects (after a short time) were revealed. Additionally, the presence of graphene oxide (GO) (5% and 7.5%) conferred relevant antibacterial features to PCL fibrous scaffolds against both *S. epidermidis* and *E. coli* by direct action on the bacterial cell membrane; meanwhile, GO-scaffolds demonstrated high biocompatibility on human foreskin fibroblasts [135]. Moreover, Tardajoos and colleagues (2018) [136] prepared 3D PCL constructs, fibers, and scaffolds with added chitosan; they observed, after 24 h of incubation, antistaphylococcal properties and high fibroblast viability. Finally, in a study by Motealleh et al. (2013) [137], chamomile-loaded wound dressing mats based on PCL were prepared; antibacterial and antifungal activities, and favorable cell adhesion/proliferation were highlighted.

The design and development of novel biomaterials with dual functions both antimicrobial and osteoinductive properties for orthopedic application are highly desirable. In fact, the integration of antimicrobial properties into implants could reduce the healing and treatment time, as well as the total costs [24,138]. Other classes of biomaterials with a broad spectrum of antimicrobial activity have been taken into consideration in the literature, but have not yet been fully explored. These include the following: polyphenol-based polymers containing functional groups such as gallol and catechol [139], biohybrids constituted via the combination of polymers and antimicrobial peptides [140], a co-delivery system that involves collagen scaffolds which are able to release both antimicrobials and bone morphogenic proteins [141], and hydrogels utilizing poly(hexamethylene guanidine) and poly(ethylene glycol) for a broad-spectrum and strong antibacterial activity [142].

## 5. Conclusions

In recent decades, research has moved into the design of advanced biomaterials that can interact with the biological environment and stimulate specific biological responses for orthopedic applications. The biomaterials used in orthopedics started as inert materials—such as titanium, titanium alloys and UHWMPE—used to restore the function and structure of hard tissue such as bone and cartilage. Thereafter, they have progressed to regenerative medicine, mainly based on scaffolds of PCL, CaPs, and composites, which play a key role in avoiding traumatic tissue damage in the earlier stages of trauma. The knowledge and tuning of the implant immune response will additionally lead to the superior incorporation of biomaterials into host tissue or the near-perfect restoration of host tissue. The possibility to change and control material properties represents a major innovation, opening a completely new range of approaches in pursuit of the desired material interaction with the biological surroundings.

Relevant and recent literature has reported that the addition of metal ions (e.g., silver, copper, etc.), natural antimicrobial compounds (e.g., essential oils, graphene, chitosan), or antioxidant agents (e.g., vitamin E) to different biomaterials conferred strong antibacterial properties and anti-adhesive features, improving their capability to counteract PJIs and biofilm formation, which remain important issues in orthopedic surgery. In addition, those biomaterials seemed to result in cytocompatible materials that supported bone self-healing when in direct contact with human osteoblast progenitor cells, leading to a suitable interplay between biocompatibility/bioactivity and the antimicrobial effect.

In conclusion, there is still refined material science to be explored. The task of adapting biomaterial structures to allow implant integration and tissue regeneration while avoiding microbial contamination seems to be an achievable challenge that could be overcome through the close interdisciplinary work of material scientists, engineers, microbiologists, chemists, physicists, and orthopedic surgeons.

## Figures and Tables

**Figure 1 antibiotics-11-00529-f001:**
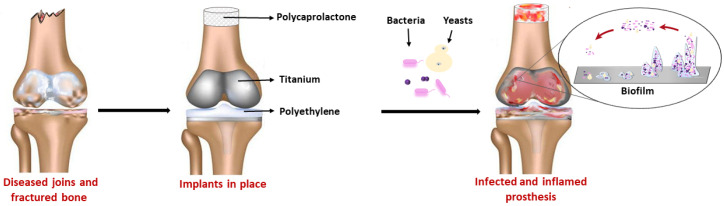
Orthopedic prosthetic surgery for the functional restoration of bone and joints is achieved surgically in total joint arthroplasties by using permanently implanted biomaterials such as titanium and polyethylene. Additionally, numerous bone fractures and other musculoskeletal problems need to be addressed using biodegradable biomaterials, such as poly(ε-caprolactone). The presence of a foreign body is a triggering event for prosthetic joint infection because the implant surface attracts free-floating bacteria, and this bacterium–surface interaction becomes irreversible thanks to biofilm formation.

**Figure 2 antibiotics-11-00529-f002:**
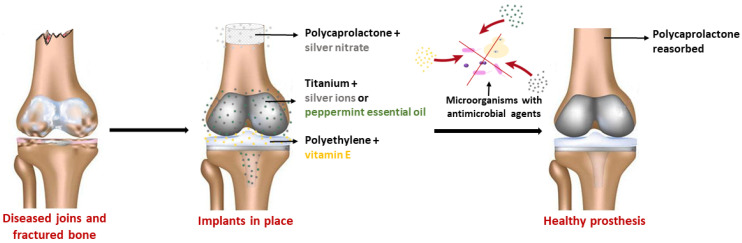
The development of biomaterials that incorporate antimicrobial agents—e.g., silver nitrate, peppermint essential oil, silver ions, and vitamin E—can reduce microbial adhesion (via contact-killing or release-killing modes, or both) on these surfaces, preventing implant-related infection.

**Table 1 antibiotics-11-00529-t001:** Summary of the materials and their orthopedic applications reported in the literature.

Material	Orthopedic Application	References
Ti and Ti alloys	Prosthesis stems; Fixation pins.	[8,16,27,28,30,31,49,50]
UHWMPE	Liner of acetabular cups in total hip arthroplasties; Tibial insert and patellar components in total knee arthroplasties; Artificial joints in shoulder arthroplasties; Spacer in intervertebral artificial disc replacement.	[8,27,32,33,34,35,36,37,38,39,40,51]
PCL+CaPs PGA+CaPs	Bioabsorbable devices for bone regeneration and drug delivery; Scaffolds for tissue engineering of cartilage and bone; Fibers for bone grafting.	[41,42,43,44,46,47,48,52,53,54,55,56,57]
PGO	Fixation pins and biodegradable fixation pins; Implants for arthrodesis; Scaffolds/membranes for tissue engineering of cartilage and bone.	[58,59,60,61]
PLA	Biodegradable devices for bone regeneration and delivery of growth factors; Scaffolds for tissue engineering of bone; Fibrous membranes guiding bone regeneration.	[42,62,63,64]

**Table 2 antibiotics-11-00529-t002:** Summary of antimicrobial and anti-adhesive compounds added to biomaterials used in orthopedic applications: data on their antimicrobial properties.

Antimicrobial Compounds	Antimicrobial and Anti-Adhesive Properties	References
Silver	Anti-adhesive and antibiofilm properties against *S. aureus* strain (ATCC 29213)	[99]
	Anti-adhesive and antibiofilm properties against *S. aureus* (ATCC 43300)	[16]
	Antibacterial effect against *E. coli* (CICC23657) and *S. aureus* (CICC10384)	[52]
	Anti-adhesive and antibacterial properties against *S. aureus* (ATCC 29213)	[53]
	Anti-adhesive and antibacterial properties against *S. aureus* (ATCC 29213)	[46]
	Antibacterial activity against *E. coli*, *P. aeruginosa*, *S. aureus*, and *C. albicans*	[100]
	Antibacterial effect against *E. coli* (BCRC 11634) and *S. aureus* (BCRC 10451)	[101]
	Antibacterial activity against *S. aureus* (Xen36) and *P. aeruginosa* (PA01)	[102]
	Anti-adhesive action on *S. aureus*, *S. epidermidis*, *Enterococcus faecalis*, *Enterobacter cloacae*, and *P. aeruginosa*	[103]
	Antibiofilm properties against *S. aureus* (ATCC25923) and *Streptococcus mutans* (UA159)	[104]
MgO–Ag nanocomposites	Antibacterial effect against *E. coli* and *S. aureus*	[73]
Diamond-like carbon coating plus silver or copper	Reduction of *E. coli* adhesion and biofilm formation; antibacterial effect against *E. coli*	[75]
Diamond-like carbon coating	Reduction of adhesion of *E. coli* (WTF 1693) and *P. aeruginosa* (ATCC 33347)	[30]
ZnO nanoparticles	Reduction of MRSA adhesion and proliferation	[47]
Essential oils (Mentha piperita)	Reduction of *S. aureus* (ATCC 29213) adhesion and biofilm formation	[105]
	Antibacterial effect on *S. aureus* ATCC 25923 and on *P. aeruginosa* ATCC 27853, and reduction of the capacity of the bacterial strains to adhere and generate biofilm. Anti-adhesive and antibiofilm properties against *C. albicans* (ATCC 10231)	[85]
	Antibacterial effect on *E. coli* (ATCC 25922 and C5), *P. aeruginosa* (ATCC 27853), *S. aureus* (MRSA 388, ATCC 25923 and ATCC 6538), *E. faecium* (DSM 13590), and on fungal strain of *C. parapsilosis* (ATCC 22019).	[86]
Vitamin E	Reduction of *S. epidermidis* (ATCC 35984)*, S. aureus* (ATCC 15981) adhesion. Variable results for the clinical strains tested	[106]
	Reduction of *S. epidermidis *(ATCC 35984, 12228 and a clinical biofilm-producing strain) adhesion and biofilm formation	[107]
	Reduction of *S. aureus* (ATCC 29213) and *E. coli* (ATCC 25922) adhesion	[108]
	Reduction of *S. epidermidis* (ATCC 35984)*, S. aureus* (ATCC 29213)*, E. coli* (ATCC 25922), and *C. albicans* (ATCC 10231) adhesion and biofilm formation	[109]
	Anti-adhesive, antibacterial, and antibiofilm properties against *S. aureus*, *S. epidermidis*, and *P. aeruginosa*	[110]

**Table 3 antibiotics-11-00529-t003:** Summary of antimicrobial and anti-adhesive compounds added to biomaterials used in orthopedic applications: data on their cytotoxicity.

Antimicrobial Compounds	Biocompatibility	References
Silver	Ag^+^ ions did not affect cytocompatibility towards osteoblast (hFOB 1.19, ATCC CRL-11372)	[16]
	Rat mesenchymal stem cells and mouse 3T3 fibroblasts did not display reduced viability in silver’s presence	[52]
	The addition of silver revealed a reduction in osteoblast-like human cells (SaOS-2, ATCC HTB-85) viability	[46]
	Human dermal fibroblasts (HDF, ATCC PCS-201-030) were viable and maintained the proper morphology at lowest Ag^+^ concentrations	[100]
	NIH/3T3 mouse embryonic fibroblasts (ATCC-CRL1658) showed good cell compatibility and low levels of cytotoxicity	[101]
	MC3T3-E1 pre-osteoblasts were affected into cell area, length, width, and fluorescence intensity in Ag presence	[104]
MgO–Ag nanocomposites	Osteoblast-like human cells (SaOS-2) were able to proliferate and differentiate (at low Ag+ concentrations)	[73]
Diamond-like carbon coating plus silver or copper	Osteoblast-like human cells (SaOS-2, ATCC) showed high proliferation levels (but depending on Ag/Cu dose) and human endothelial cells (EA.hy926, ATCC) showed depletion in viability	[75]
ZnO nanoparticles	Human mesenchymal stem cells were supported in osteodifferentiation	[47]
Essential oils (Mentha piperita)	Human MG-63 cell line (ATCC CRL-1427) had highest proliferation rates in samples with EOs compared to the control	[85]
Vitamin E	Improved the ability of SaOS-2 to respond to oxidative stress	[111]

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
