# Peer review of "Current Knowledge on Biomaterials for Orthopedic Applications Modified to Reduce Bacterial Adhesive Ability"

_antibiotics, 2022, doi:10.3390/antibiotics11040529_

Round 1
Reviewer 1 Report
From the microbiological point of view, the manuscript is appropriate redacted, the authors results are compared with similar studies from the literature.
I have only a few suggestions that might improve the quality of the manuscript:
- the authors should revise the use of acronyms in the manuscript, there are a lot of acronyms in very different part of the manuscript, forcing the reader to return to each first acronym, multiple times
- the authors should complete the table 2 with the antibacterial activity of silver against different bacterial strains, not only on S. aureus
- the authors should consider to discuss a few others potential solutions for the antimicrobial activity that might be taken into consideration for their antimicrobial activity (biosynthesized nanoparticles, etc.)
Author Response
REVIEWER 1
From the microbiological point of view, the manuscript is appropriate redacted, the authors results are compared with similar studies from the literature.
I have only a few suggestions that might improve the quality of the manuscript:
- the authors should revise the use of acronyms in the manuscript, there are a lot of acronyms in very different part of the manuscript, forcing the reader to return to each first acronym, multiple times
REPLY: Done, in agreement with reviewer suggestion, we revised the whole manuscript for acronyms and added, at the end of the paper, the section “Acronyms”.
- the authors should complete the table 2 with the antibacterial activity of silver against different bacterial strains, not only on S. aureus
REPLY: Done, in agreement with reviewer suggestion, we added in Table 2 the antibacterial activity of silver on other bacteria than Staphylococcus aureus, including Gram-positive and Gram-negative strains.
- the authors should consider to discuss a few others potential solutions for the antimicrobial activity that might be taken into consideration for their antimicrobial activity (biosynthesized nanoparticles, etc.)
REPLY: Done, in agreement with reviewer suggestion, we added other antibacterial solutions on biomaterials in the latest part of Section 4.
Reviewer 2 Report
Should not be published in Antibiotics
In their review article entitled “Biomaterials Modified to Reduce Bacterial Adhesive Ability: Current Knowledge on Counteracting Biomaterial-Associated Infections in Orthopedics”, Valeria Allizond et al. make a state of the art structured around their own work done during the last 10 years. This review first presents (i) some biomaterials used in orthopedics for replacement or regeneration purposes and then highlights (ii) a selection of some antibacterial and anti-adhesive compounds. The last part of this review describes how the abovementioned components have been used to design innovative biomaterials with anti-adhesive and antimicrobial properties. Such developments done by the authors are discussed with regard to some other related works recently reported in the literature.
General appreciation
This manuscript is based on 117 references to previous studies; with respect to the year of publication, the mean and median are 2017 +/- 4.1 and 2018, respectively, and 54% of the studies considered were published less than 5 years ago (i.e. after 2017). From this point of view, this review looks fine. However, several comments must be done. First, other reviews, more extensive than the present one, have been published quite recently (see below). Second, the title of the manuscript is misleading since it suggests an overview of the literature, which is clearly not the case. Third, the general structure of this paper is questionable. Authors focused on their own works and other related ones and by doing so they excluded many other works exploring other strategies. This makes this paper resembling an opinion paper rather than a classical – strictly speaking – review article. The main part of the review (that is in line with the title of the paper) is the last one, which accounts for less than half of the length of the paper. Overall, this structure looks inappropriate and would require to be thoroughly revised (read below). Considering altogether the important issues noted, I have to recommend rejection. Below are listed more specific comments that the authors should consider for improving their manuscript in view of a resubmission.
Main comments
The title must be modified to avoid disappointment and better describe the content of the paper.
The paper should be revised to clearly highlight that the options presented are selections, stating that it is not intended to be exhaustive, and referring to previous excellent more detailed reviews (see below).
The overall organization of the paper must be reworked, with a clearly defined topic for each section. Section 2 provides some information that are repeated in Sections 3 and 4 (e.g. the paragraph lines 222-228). Section 3 should be merged with the last part (current section 4), which should contain subsections.
The paper contains a single Figure. It is mentioned only once, at the very end of the manuscript; however, it could serve in other instances in the paper. Another figure(s) would be helpful for illustrative purposes.
As for references, others should be considered by the authors, such as:
Afewerki S, Bassous N, Harb S, Palo-Nieto C, Ruiz-Esparza GU, Marciano FR, Webster TJ, Furtado ASA, Lobo AO. Advances in dual functional antimicrobial and osteoinductive biomaterials for orthopaedic applications. Nanomedicine. 2020 Feb;24:102143. https://doi.org/10.1016/j.nano.2019.102143
Lu H, Liu Y, Guo J, Wu H, Wang J, Wu G. Biomaterials with antibacterial and osteoinductive properties to repair infected bone defects. Int J Mol Sci 2016;17(3):334, https://doi.org/10.3390/ijms17030334
Raphel J, Holodniy M, Goodman SB, Heilshorn SC. Multifunctional coatings to simultaneously promote osseointegration and prevent infection of orthopaedic implants. Biomaterials 2016;84:301-14, https://doi.org/10.1016/j.biomaterials.2016.01.016
Winkler H. Treatment of chronic orthopaedic infection. EFORT Open Revi 2017;2(5):110-6, https://doi.org/10.1302/2058-5241.2.160063
Ribeiro M, Monteiro FJ, Ferraz MP. Infection of orthopedic implants with emphasis on bacterial adhesion process and techniques used in studying bacterial-material interactions. Biomatter 2012;2(4):176-94, https://doi.org/10.4161/biom.22905
Lin X, Yang S, Lai K, Yang H, Webster TJ, Yang L. Orthopedic implant biomaterials with both osteogenic and anti-infection capacities and associated in vivo evaluation methods. Nanomedicine 2017;13 (1):123-42, https://doi.org/10.1016/j.nano.2016.08.003
Fernandes JS, Gentile P, Pires RA, Reis RL, Hatton PV. Multifunctional bioactive glass and glass-ceramic biomaterials with antibacterial properties for repair and regeneration of bone tissue. Acta Biomater 2017;59:2-11, https://doi.org/10.1016/j.actbio.2017.06.046
Im GI. Biomaterials in orthopaedics: the past and future with immune modulation. Biomater Res. 2020 Feb 4;24:7. https://doi.org/10.1186/s40824-020-0185-7
Other comments
The paper contains a section 2.2.1 but no section 2.2.2.
Some sentences are very lengthy and/or not clear and should be rephrased.
In Figure 1, some items are not explicitly defined (e.g. bacteria)
Line 62, check the sentence.
Line 36, check “healthy lifestyle is are adopted”.
The first paragraph in section 4 looks misplaced or out of topic there. The last paragraph in section 4 also looks not well inserted.
Author Response
REVIEWER 2
Should not be published in Antibiotics
In their review article entitled “Biomaterials Modified to Reduce Bacterial Adhesive Ability: Current Knowledge on Counteracting Biomaterial-Associated Infections in Orthopedics”, Valeria Allizond et al. make a state of the art structured around their own work done during the last 10 years. This review first presents (i) some biomaterials used in orthopedics for replacement or regeneration purposes and then highlights (ii) a selection of some antibacterial and anti-adhesive compounds. The last part of this review describes how the abovementioned components have been used to design innovative biomaterials with anti-adhesive and antimicrobial properties. Such developments done by the authors are discussed with regard to some other related works recently reported in the literature.
General appreciation
This manuscript is based on 117 references to previous studies; with respect to the year of publication, the mean and median are 2017 +/- 4.1 and 2018, respectively, and 54% of the studies considered were published less than 5 years ago (i.e. after 2017). From this point of view, this review looks fine. However, several comments must be done. First, other reviews, more extensive than the present one, have been published quite recently (see below). Second, the title of the manuscript is misleading since it suggests an overview of the literature, which is clearly not the case. Third, the general structure of this paper is questionable. Authors focused on their own works and other related ones and by doing so they excluded many other works exploring other strategies. This makes this paper resembling an opinion paper rather than a classical – strictly speaking – review article. The main part of the review (that is in line with the title of the paper) is the last one, which accounts for less than half of the length of the paper. Overall, this structure looks inappropriate and would require to be thoroughly revised (read below). Considering altogether the important issues noted, I have to recommend rejection. Below are listed more specific comments that the authors should consider for improving their manuscript in view of a resubmission.
REPLY: We uploaded a revised version of the manuscript, now we hope that the reviewer will consider it suitable for publication on Antibiotics. It was difficult to address all the reviewer’s concerns since strong issues were raised, anyway we did our best to improve the overall manuscript quality.
Main comments
The title must be modified to avoid disappointment and better describe the content of the paper.
REPLY: Done, as suggested by the reviewer, we modified the title of the manuscript.
The paper should be revised to clearly highlight that the options presented are selections, stating that it is not intended to be exhaustive, and referring to previous excellent more detailed reviews (see below).The overall organization of the paper must be reworked, with a clearly defined topic for each section. Section 2 provides some information that are repeated in Sections 3 and 4 (e.g. the paragraph lines 222-228). Section 3 should be merged with the last part (current section 4), which should contain subsections.
REPLY: Done, in agreement with reviewer suggestion, we rewrite the section 4, but we decided to not merge section 3 and 4, since no other reviewers recommended this option.
The paper contains a single Figure. It is mentioned only once, at the very end of the manuscript; however, it could serve in other instances in the paper. Another figure(s) would be helpful for illustrative purposes.
REPLY: Done, in agreement with reviewer suggestion, we split the Figure into Figure 1 and Figure 2 to better explain the concepts of orthopaedic prosthetic surgery for functional restoration and the development of biomaterials functionalized antimicrobials to counteract microbial infection, respectively.
As for references, others should be considered by the authors, such as:
Afewerki S, Bassous N, Harb S, Palo-Nieto C, Ruiz-Esparza GU, Marciano FR, Webster TJ, Furtado ASA, Lobo AO. Advances in dual functional antimicrobial and osteoinductive biomaterials for orthopaedic applications. Nanomedicine. 2020 Feb;24:102143. https://doi.org/10.1016/j.nano.2019.102143
Lu H, Liu Y, Guo J, Wu H, Wang J, Wu G. Biomaterials with antibacterial and osteoinductive properties to repair infected bone defects. Int J Mol Sci 2016;17(3):334, https://doi.org/10.3390/ijms17030334
Raphel J, Holodniy M, Goodman SB, Heilshorn SC. Multifunctional coatings to simultaneously promote osseointegration and prevent infection of orthopaedic implants. Biomaterials 2016;84:301-14, https://doi.org/10.1016/j.biomaterials.2016.01.016
Winkler H. Treatment of chronic orthopaedic infection. EFORT Open Revi 2017;2(5):110-6, https://doi.org/10.1302/2058-5241.2.160063
Ribeiro M, Monteiro FJ, Ferraz MP. Infection of orthopedic implants with emphasis on bacterial adhesion process and techniques used in studying bacterial-material interactions. Biomatter 2012;2(4):176-94, https://doi.org/10.4161/biom.22905
Lin X, Yang S, Lai K, Yang H, Webster TJ, Yang L. Orthopedic implant biomaterials with both osteogenic and anti-infection capacities and associated in vivo evaluation methods. Nanomedicine 2017;13 (1):123-42, https://doi.org/10.1016/j.nano.2016.08.003
Fernandes JS, Gentile P, Pires RA, Reis RL, Hatton PV. Multifunctional bioactive glass and glass-ceramic biomaterials with antibacterial properties for repair and regeneration of bone tissue. Acta Biomater 2017;59:2-11, https://doi.org/10.1016/j.actbio.2017.06.046
Im GI. Biomaterials in orthopaedics: the past and future with immune modulation. Biomater Res. 2020 Feb 4;24:7. https://doi.org/10.1186/s40824-020-0185-7
REPLY: Done, in agreement with reviewer suggestion, we added the proposed references within the manuscript.
Other comments
The paper contains a section 2.2.1 but no section 2.2.2.
REPLY: Done, in agreement with reviewer suggestion, we divided the section 2.2.1 in two different ones: 2.2.1 (Polyε-caprolactone) and 2.2.2 (Calcium phosphates and composites).
Some sentences are very lengthy and/or not clear and should be rephrased.
REPLY: Before the submission, the manusctipt was edited for the english by the MPDI english editing service, we are surprised that several phrases/concepts not clear still remained.
In Figure 1, some items are not explicitly defined (e.g. bacteria)
REPLY: DONE
Line 62, check the sentence.
REPLY: DONE
Line 36, check “healthy lifestyle is are adopted”.
REPLY: DONE
The first paragraph in section 4 looks misplaced or out of topic there. The last paragraph in section 4 also looks not well inserted.
REPLY: Done, in agreement with reviewer suggestion, we replaced and rephrased the two sentences.
Reviewer 3 Report
The authors also add the applications of PGA, PLA and PDO in the table 1
Author Response
REVIEWER 3
The authors also add the applications of PGA, PLA and PDO in the table 1
REPLY: Done, in agreement with reviewer suggestion, we added the applications of PGA, PLA and PDO in Table 1.
Reviewer 4 Report
On request of Antibiotics, I have revised the review entitled "Biomaterials modified to reduce bacterial adhesive ability: current knowledge to counteract biomaterial associate infections in orthopaedics" by Valeria Allizond and co-authors.
With this work the authors aimed to provide an systematic overview of the evolution of biomaterials in orthopedic applications while considering the properties and modification/functionalization leading to improvement in preventing microbial adhesion and biofilm formation and promoting host cell proliferation and colonization.
The present review is well written, English is fine (only a minor spell check is required), but I recommend reformulating section 4. Biomaterials coupled with anti-adhesive and antimicrobial agents. It is confusing. Biomaterials with antibacterial properties cannot be discussed as long as many of them have not been evaluated for biocompatibility or if they exert toxic effects on human cells (Table 2). Also, I recommend the authors to rename the subchapters 3.1 and 3.2 as "Metal ions" and "Essential oils".
Author Response
REVIEWER 4
On request of Antibiotics, I have revised the review entitled "Biomaterials modified to reduce bacterial adhesive ability: current knowledge to counteract biomaterial associate infections in orthopaedics" by Valeria Allizond and co-authors.
With this work the authors aimed to provide an systematic overview of the evolution of biomaterials in orthopedic applications while considering the properties and modification/functionalization leading to improvement in preventing microbial adhesion and biofilm formation and promoting host cell proliferation and colonization.
The present review is well written, English is fine (only a minor spell check is required), but I recommend reformulating section 4. Biomaterials coupled with anti-adhesive and antimicrobial agents. It is confusing.
REPLY: Done, in agreement with reviewer suggestion, we rewrite the section 4.
Biomaterials with antibacterial properties cannot be discussed as long as many of them have not been evaluated for biocompatibility or if they exert toxic effects on human cells (Table 2).
REPLY: Done, in agreement with reviewer suggestion, we divided table 2 into two different tables. In particular, Table 2 with the antibacterial results and Table 3 with the cytotoxicity data.
Also, I recommend the authors to rename the subchapters 3.1 and 3.2 as "Metal ions" and "Essential oils".
REPLY: DONE
Round 2
Reviewer 2 Report
In their revised review article now entitled “Current Knowledge on Biomaterials for Orthopedic Applications Modified to Reduce Bacterial Adhesive Ability”, Valeria Allizond et al. did noticeable modifications to their original manuscript. I appreciate all the changes done for taking into account most of the issues raised. All combined, I think these modifications increase the quality and expected impact of this manuscript, making it now suitable for publication in Antibiotics. I would only recommend doing further slight improvements: the non-exhaustive character of this review should also be mentioned in the abstract (not only in section 4); the last section may be divided in several sub-sections and some lengthy sentences may be rephrased/shortened; some typo errors should be fixed (e.g. line 129, “possesses”; line 156, “giuding”).
Author Response
REVIEWER 2
In their revised review article now entitled “Current Knowledge on Biomaterials for Orthopedic Applications Modified to Reduce Bacterial Adhesive Ability”, Valeria Allizond et al. did noticeable modifications to their original manuscript. I appreciate all the changes done for taking into account most of the issues raised. All combined, I think these modifications increase the quality and expected impact of this manuscript, making it now suitable for publication in Antibiotics.
REPLY: We thank reviewer 2 for appreciating our work in revising the manuscript to make it publishable on Antibiotics.
I would only recommend doing further slight improvements: the non-exhaustive character of this review should also be mentioned in the abstract (not only in section 4); the last section may be divided in several sub-sections and some lengthy sentences may be rephrased/shortened; some typo errors should be fixed (e.g. line 129, “possesses”; line 156, “giuding”).
REPLY: Done, in agreement with reviewer suggestion, we added the sentence into the abstract and we divided Section 4 in two different sub-sections. We checked Section 4 for lengthy sentences and the whole manuscript for typos.